# Health-Related Quality of Life (HRQoL) and the Effect on Outcome in Patients Presenting with Coronary Artery Disease and Treated with Percutaneous Coronary Intervention (PCI): Differences Noted by Sex and Age

**DOI:** 10.3390/jcm11175231

**Published:** 2022-09-05

**Authors:** Andre Conradie, John Atherton, Enayet Chowdhury, MyNgan Duong, Nisha Schwarz, Stephen Worthley, David Eccleston

**Affiliations:** 1Friendly Society Private Hospital, Bundaberg, QLD 4670, Australia; 2Royal Brisbane and Women’s Hospital, Brisbane, QLD 4029, Australia; 3GenesisCare, Leabrook, SA 5068, Australia; 4North Shore Cardiology, St Leonards, NSW 2065, Australia; 5Melbourne Private Hospital, Parkville, VIC 3052, Australia

**Keywords:** acute coronary syndrome, PCI, quality of life, clinical outcomes, sex, MACE

## Abstract

Background and aim: poor quality of life (QoL) has been identified as an independent risk factor for mortality and major cardiac events (MACE) in patients with cardiovascular disease (CVD). The aim of this study was to assess health-related quality of life (HRQoL) at baseline and its association with outcome in patients with coronary artery disease presenting for percutaneous coronary intervention (PCI). The outcome was measured by mortality and MACE at 1-year, and whether there was any difference for sex and different age groups. Methods and results: all patients prospectively enrolled into the GenesisCare Outcome Registry (GCOR) over a 11-year period were included in the study. The EQ-5D-5L and VAS patient survey were used for assessment of baseline HRQoL. Of the 15,198 patients, only 6591 (43.4%) completed the self-assessment. Women had significantly more impairment of all five dimensions of the EQ-5D-5L survey, and their self-reported QoL was significantly lower than men (68.3 in women vs. 71.9 in men, *p* < 0.001). Poor QoL was strongly associated with increased mortality (HR 2.85; 95% CI 1.76 to 4.62, *p* < 0.001) and MACE (HR 1.40; 95% CI 1.10 to 1.79, *p* = 0.01). A similar trend was noted for women and men, but did not reach significance in women due to the smaller number of female patients. Conclusion: poor HRQoL is associated with subsequent mortality and MACE in patients undergoing PCI. By not assessing quality of life as a standard of care, an opportunity is lost to identify high-risk patients who may benefit from targeted interventions to improve health outcomes.

## 1. Introduction

Cardiovascular disease (CVD) is still worldwide the major cause of death and morbidity [1]. With the improvement in care over the last three decades, a significant improvement in survival has been witnessed, resulting in more patients living with chronic CVD. CVD and its associated treatment have the potential of affecting patients on physical, social, and psychological levels, and can thus have a significant impact on health-related quality of life (HRQoL) [1,2,3]. By assessing the health status of patients with established coronary artery disease, presenting for percutaneous coronary intervention (PCI), we may identify whether some patients could benefit from targeted interventions to improve long-term clinical outcomes. The additional benefit is the potential effect on improving outcome as measured by mortality, major adverse cardiovascular events (MACE), reduced unplanned readmissions to hospital, and improved adherence to secondary prevention medication [4,5]. Recently, the focus has been more directed towards a patient-centred model of care, and the patients’ health status is considered as a vital metric to more accurately determine the effect of cardiac disease and its treatment on the quality of life of patients [6]. Self-reported health status of patients has been identified as an independent risk factor for mortality in patients with cardiovascular disease, with some studies suggesting it to be even more important than the current biomedical risk factors [5]. Multiple measures exist to determine the severity of cardiovascular disease (left ventricular function or NT-proBNP as physiological measurement of the severity of heart failure, exercise stress testing or coronary imaging to assess the burden of coronary artery disease); but very poor correlation was noted between these functional studies and how patients perceived their quality of life [6]. Assessment of the patients’ health status by clinicians has also been reported to poorly correlate with the patients’ perception of their health status, reinforcing the use of standardised patient-reported surveys as the most accurate way to reflect on patients’ HRQoL [4,5,6]. With the strong association of health status with prognosis of cardiovascular disease and effectiveness of treatment, it has been proposed that assessment of a patient’s health status should be complimentary to all the other usual modalities of assessment, and contribute as an essential measure of health and the quality of care [5,6,7]. Not only does health status impact on outcomes, but also has a direct effect on health expenditure, with patients in the lowest HRQoL range consuming three times more of the annual healthcare budget compared to patients in the higher HRQoL range. All of these factors support the incorporation of HRQoL instruments in the baseline assessment of patients presenting with cardiovascular disease [6,8].

The EQ-5D-5L health status survey is an internationally recognised generic instrument, well validated for the assessment of patients’ health status in cardiovascular disease, and has the benefit of being brief and very easy to use [9]. The aim of this study was to assess the HRQoL of patients at baseline in patients undergoing PCI, and to determine whether this correlated with subsequent outcome as measured by mortality and MACE. The correlation with subsequent myocardial infarction, target vessel revascularisation (TVR), unplanned cardiac readmissions, and adherence to all four classes of evidence-based drugs used for secondary prevention was also assessed. Finally, we evaluated whether there was a difference according to age and sex.

## 2. Methods

### 2.1. Study Design/Data Source

Data were obtained from the GenesisCare Outcome registry (GCOR) and included all patients prospectively enrolled to the database over a 11-year period from January 2009 to December 2019. Patients eligible for the study must have completed a 1-year follow-up period, with complete data available on baseline demographics and all clinical variables. An EQ-5D-5L patient survey was issued to every patient with the index PCI procedure, to be completed at discharge or to be posted back. Patients were supplied with a patient information sheet and an opt-out consent was utilised.

### 2.2. Ethical Approval

The PCI Registry and its entirety have been approved by the Bellberry Human Research Ethics Committee (HREC).

### 2.3. Health Status

The EQ-5D is a generic health instrument developed by the EuroQol Group [10,11] and has been validated for use in patients with cardiovascular disease (including coronary artery disease and heart failure) as well as in the general population [9,12]. The EQ-5D-5L consist of two components, a descriptive system, and a visual analogue scale (VAS). The descriptive system comprises five components assessing mobility, personal care, usual activity, pain/discomfort, and anxiety/depression, with all five dimensions assessed on five levels: having no problems, slight problems, moderate problems, severe problems, and extreme problems. The visual analogue scale (VAS) is a patient self-rated system ranging from 0 (worse imaginable health state) to 100 (best imaginable health state). For assessment of the five dimensions and their potential effect on outcome, we dichotomise the possible responses in to 0 (having no problem) and 1 (having any problem ranging from slight to extreme), and for the visual analogue scale (VAS) the cut-off point of ≤60 was used as poor self-rated health status [12].

### 2.4. Baseline Measures, Follow-Up, and Clinical Outcomes

Socio-demographic parameters, medical history and management including in-hospital investigations and procedure details were routinely recorded. All in-hospital complications following PCI were recorded at the time of discharge. For the current study, follow-up information for one-year post-procedure was used. Follow-up assessments were performed by research coordinators at discharge, at 30-day, and at 1-year post-procedure. All cardiac events were documented following review of medical records including death, myocardial infarction (MI), target vessel revascularisation (TVR, defined as revascularisation of a previously treated artery), and rehospitalisation. MACE was the composite of death, MI and/or TVR events. Secondary prevention medication included adherence to statins, angiotensin-converting enzyme inhibitors (ACEI)/angiotensin receptor blockers (ARB), beta-blockers, and antiplatelet therapy (aspirin/clopidogrel/ticagrelor/prasugrel).

### 2.5. Statistical Methods

Patients in the PCI registry who completed the EQ-5L-5D questionnaire at baseline and had complete covariate information available were included for the current analysis. A descriptive analysis was used (frequency/sample proportions or mean with standard deviations) to summarise baseline characteristics (e.g., age, clinical parameters, etc.), and risk factors of the patients by availability of data. Student t-test, ANOVA or chi-square tests were used to compare the distributions of baseline characteristics including risk factors by availability of data. Among those with data available, a comparison was made of the baseline characteristics by baseline/pre-procedure QoL status based on VAS (i.e., poor versus good). The distribution of clinical outcomes by presence of problem was explored among those with QoL information available. The distribution of outcomes by sex and age for different responses was also explored. Further, the association of clinical outcomes within one-year post-procedure was explored with pre-procedural QoL status using univariate and multivariate Cox proportional hazard regression models. Multivariate regression models were adjusted for all confounding variables with a significance of *p* < 0.05 between good and poor QoL, including patient age, sex, body mass index (BMI), smoking status, PCI presentation (i.e., STEMI/Non-STEMI) at baseline, diabetes, left ventricular ejection fraction (LVEF), previous myocardial infarction (MI), peripheral vascular disease (PVD), previous coronary artery bypass surgery (CABG), heart failure within prior 2 weeks, renal failure, and cardiogenic shock. All statistical analyses were performed using Stata version 15.2 for Windows (StataCorp LP, College Station, TX, USA).

## 3. Results

### 3.1. Clinical Characteristics and Differences between Patients Completing and Not Completing QoL Surveys

Of the 15,198 patients enrolled in the GCOR PCI database, 6591 (43.4%) completed the EQ-5L-5D questionnaire. Appendix A summarises the baseline characteristics of the patients with a completed EQ-5L-5D and those who did not. A similar distribution was noted for most of the baseline characteristics between those with and those without QoL information available except for age, family history of coronary artery disease, history of smoking (past or current), body mass index (BMI), history of previous PCI, current heart failure, and a history of renal failure (serum creatinine > 2 mg/dL and/or receiving dialysis). The ratio between men and women was similar in both groups (*p* = 0.19), with patients completing the QoL survey significantly older (69.2 ± 10.0 years completing the survey vs. 68.4 ± 10.9 years not completing survey, *p* < 0.001). Patients presenting for an elective procedure (stable angina with a positive stress test) had a higher rate of completing the survey (56.7% completing survey vs. 52.5% not completing survey, *p* < 0.001), while a persistently lower completion rate was noted in patients when presenting with a STEMI or cardiogenic shock.

### 3.2. Baseline Characteristics of Patients by Self-Assessed HRQoL Status and Differences Observed for Sex and Age

With the assessment of patients’ self-rated quality of life (Table 1), 5223 patients completed the VAS, with 1055 (20%) of the patients assessing their general health to be of very poor quality (≤60). Significant differences were noted between patients with good QoL compared to those who perceived their quality of life as very poor. Patients who perceived their general health as poor included a higher proportion of women and were significantly younger (68.0 ± 10.9 years. with poor QoL vs. 69.2 ± 9.7 years. with good QoL, *p* < 0.001). In patients with a poor QoL a higher prevalence of diabetes (26.9% poor QoL vs. 22.9% good QoL, *p* = 0.01) was noted, with BMI significantly higher (29.6 ± 5.9 poor QoL vs. 28.8 ± 4.8 good QoL, *p* < 0.001), and left ventricular ejection fraction (55.3 ± 10.0 poor QoL vs. 57.1 ± 9.2 good QoL, *p* < 0.001) significantly lower. Patients with a poor quality of life had a higher rate of established coronary artery disease, with a higher proportion having a history of previous myocardial infarction (24.5% poor QoL vs. 20.9% good QoL, *p* = 0.01), and previous CABG (12.6% poor QoL vs. 9.5% good QoL, *p* = 0.003). Involvement of other vascular territories were also more common in patients reporting poor QoL, with a significantly higher prevalence of peripheral vascular disease (9.3% poor QoL vs. 6.4% good QoL, *p* < 0.001). A history of previous cerebro-vascular disease (8.1% poor QoL vs. 6.8% good QoL, *p* = 0.15) was also more common in patients with poor QoL but did not reach significance. The diagnosis of heart failure was more frequent in patients with a poor quality of life, including both a previous history of heart failure (6.4% poor QoL vs. 4.9% good QoL, *p* = 0.05) as well as newly diagnosed heart failure (4.9% poor QoL vs. 2.5%, *p* < 0.001). The presence of renal failure (6.7% vs. 3.8%, *p* < 0.001) was also significantly higher in patients with poor quality of life. Patients with poor quality of life were more likely to present with STEMI (10.2% poor QoL vs. 4.8% good QoL, *p* < 0.001) and NSTEMI (29.0% poor QoL vs. 19.8% good QoL, *p* < 0.001), and had a lower probability of presenting with stable coronary artery disease (43.8% poor QoL vs. 59.3% good QoL, *p* < 0.001). The percentage of patients presenting with multivessel disease were similar for both groups (*p* = 0.21).

In patients who completed the survey, a significant difference was noted between women and men for both the five dimensions and VAS score (Table 2). Women had a persistently higher impairment rate for all five dimensions, including mobility, personal care, usual activity, pain/discomfort, and anxiety/depression (*p* < 0.001). The quality of life as assessed by the VAS score was also significantly lower in women compared to men (67.7 in women vs. 71.7 in men, *p* < 0.001).

The effect of age on quality of life was assessed by dividing the patients into three age groups: <55 years, 55–74 years, and ≥75 years (Table 3). A statistically significant difference was noted, with older patients presenting with a higher level of impairment for mobility, personal care, and usual activity (*p* < 0.001). When testing for significance of trend, all three dimensions achieved statistical significance. When assessing for pain/discomfort, older patients had a higher level of impairment (*p* < 0.001) but significance for trend was not as strong as seen with the modalities of physical activity (*p* = 0.19 for pain/discomfort). Younger patients reported a higher level of anxiety/depression (*p* < 0.001 for anxiety/depression), with significance for trend (*p* = 0.01). When assessing the VAS score, younger patients (<55 years) perceived their quality of life as significantly worse compared to older patients (*p* < 0.001) but did not reach significance when testing for trend (*p* = 0.54).

### 3.3. Descriptive Components of EQ-5D and VAS Score and Effect on Outcome

In the cohort of patients in whom complete data on QoL were available, the five modalities of the descriptive component of the EQ-5D-5L (Table 4) were significantly associated with outcome. Any problem with mobility was significantly associated with an increased risk of death (*p* = 0.01), MACE, MI, and unplanned cardiac readmissions (*p* < 0.001). Impairment of personal care was not associated with outcome, although a trend towards significance was noted for increased risk of death (*p* = 0.06). A decrease in usual activity was significantly associated with an increase in MACE (*p* = 0.03), MI (*p* = 0.003), and unplanned cardiac readmissions (*p* < 0.001). The domain of pain/discomfort was associated with an increased risk of MACE (*p* = 0.01), MI (*p* = 0.01), and unplanned cardiac readmissions (*p* = 0.01). No association was noted between anxiety/depression and risk of events. Involvement of all five modalities were very strongly associated with an increased risk of death (*p* = 0.03), with no effect on the risk of MACE, MI, or unplanned cardiac readmissions.

Patients who assessed their QoL as poor (Table 5) had a higher mortality (2.4% poor Qol vs. 0.8% good QoL, *p* < 0.001) as well as a higher rate of MACE (7.9% poor QoL vs. 6.1% good QoL, *p* = 0.04) and MI (5.5% poor QoL vs. 4.1% good QoL, *p* = 0.047). No association was noted between QoL and adherence to secondary prevention medication, attending cardiac rehabilitation or unplanned cardiac readmissions. When assessing for the association between poor quality of life and outcomes, no association was demonstrated in female patients, except for poorer attendance at cardiac rehabilitation in women (38.4% poor QoL vs. 46.8% good QoL, *p* = 0.02). In men, poor QoL was significantly associated with an increased risk of death (2.4% poor QoL vs. 0.5% good QoL, *p* < 0.001), but no association was demonstrated with MACE, MI, or unplanned cardiac readmission.

The association of baseline HRQoL with death and MACE at 1-year were assessed by Kaplan-Meier curves and adjusted Cox regression models. All patients with missing information on covariates were excluded for both models. With univariable Cox regression analyses (Appendix A) poor QoL was associated with a significant increase in risk of death (HR 3.09; 95% CI 1.89 to 5.06, *p* < 0.001) and MACE (HR 1.36; 95% CI 1.06 to 1–74, *p* = 0.02). Further, when multivariable Cox regression analyses was performed, adjusting for all confounding variables with a significance of *p* < 0.05 between good and poor QoL, the increased risk persisted. The adjusted risk of death (Figure 1) in patients who reported poor quality of life was significantly elevated (HR 2.41; 95% CI 1.44 to 4.02, *p* = 0.001). When adjusted for clinical variables, poor quality of life was associated with a significantly increased risk of death in men (HR 3.99; 95% CI 2.09 to 7.65, *p* < 0.001). A similar trend of increased risk was noted in women (HR 1.02; 95% CI 0.41 to 2.54, *p* = 0.97) but did not reach clinical significance due to the smaller number of female patients. The adjusted risk of MACE (Figure 2) in patients who reported poor quality of life was significantly higher (HR 1.31; 95% CI 1.02 to 1.69, *p* = 0.04). Poor QoL in both men (HR 1.31; CI 0.97 to 1.77, *p* = 0.07) and women (HR 1.27; CI 0.78 to 2.08, *p* = 0.34) was associated with a trend towards increased risk of MACE but did not reach significance.

## 4. Discussion

A very low completion rate of the EQ-5D-5L and VAS questionnaire was observed in this study. In patients on whom information was available, poor HRQoL was more common in women and in younger patients, and strongly correlated with increased mortality and MACE at 1-year. Furthermore, significant predictors of poor HRQoL at baseline were identified.

There is growing recognition that pre-existing health status has the potential to impact clinical outcomes as measured by mortality, risk of myocardial infarctions, and unplanned readmissions [13,14]. This may occur either directly, or indirectly, through the effect of quality of life on treatment adherence, including life-style modification programs (e.g., cardiac rehabilitation, smoking cessation programs) and medication adherence. The health status of patients can be significantly affected by multiple factors including psychological factors, socio-economic status, and ageing, with evidence emerging over the last few decades that pre-existing health-related quality of life can have an independent effect on outcome in patients with coronary artery disease when treated with PCI [5,12]. Indeed, this potential effect appears to be independent of traditional risk factors and comorbidities [4,15].

It is clear from the literature that the use of patient-reported health status is underutilised [5,14]. In the GCOR registry only 43.4% of patients completed their EQ-5D survey forms, with younger patients and patients presenting with an acute coronary syndrome having a lower rate of return. Other factors associated with lower completion rates include a positive family history for coronary artery disease, a history of previous PCI, heart failure (newly diagnosed or established) and cardiogenic shock. Indeed, given the low completion rates we observed as part of a structured program within a registry, it is likely that quality of life is not captured in most patients undergoing PCI. A possible contributing factor could be that the health status of a patient is still not seen as part of the standard clinical assessment [5]. It is also possible that the mental status of patients could have contributed to the low completion rates, with the incidence of depression reported to be as high as 45% in patients presenting with an acute coronary syndrome [16]. In the early stage of the GCOR-PCI registry the self-assessment questionnaires were solely patient dependent, and it was only after clinical processes were improved that a steady increase in self-assessment were noticed.

### 4.1. Potential Clinical Correlates of Pre-Existing Poor QoL in Patients Presenting for PCI

The baseline health status of a patient is not only affected by the presenting disease, but also by additional pre-existing status [7,17]. In this study factors associated with a poor QoL at baseline included a diagnosis of diabetes, established cardiovascular disease, previous myocardial infarction, heart failure, CKD, previous CABG, increased BMI, reduced left ventricular ejection fraction, and if the clinical presentation was that of an acute coronary syndrome.

A history of previous myocardial infarction has been well established to have a lasting effect on HRQoL [18,19] and is in line with this study. Very limited research and conflicting results are available on the type of acute coronary syndrome and its effect on quality of life in patients. In a study by Uchmanowicz et al. patients presenting with NSTEMI perceived their quality of life as much worse than patients presenting with STEMI [20]. A study by Yuval and co-workers [21] however failed to demonstrate any difference, with patients experiencing their hospitalisation equally traumatic regardless of the type of acute coronary syndrome. In this analysis of the GCOR registry, patients presenting with either NSTEMI or STEMI perceived their quality of life as significantly impacted by the acute event, compared to patients presenting for elective PCI with a diagnosis of stable coronary artery disease.

Both diabetes and high BMI were very strongly associated with poor baseline quality of life in the GCOR registry. This was in line with multiple studies investigating the impact of diabetes on the health status of patients presenting with ACS and treated with PCI [7], confirming a negative effect on health status not only at baseline [20] but also in the long-term after presenting with an acute coronary syndrome [22]. The association of diabetes with a poor baseline health status was most likely driven by the fact that the diabetic patient cohort was on average older, had more comorbidities, and had a higher burden of pre-existing cardiovascular disease including multivessel coronary artery disease [20]. In the GCOR registry increased BMI was strongly associated with poor quality of life. This is supported in the literature with obesity correlating with both a lower EQ-5D-5L index score [23] and VAS score [24]. The association of obesity with a decrease in HRQoL is furthermore demonstrated to be independent from co-morbidities associated with obesity [24].

In this study heart failure was significantly associated with poor quality of life at baseline and is in concordance with large studies demonstrating both heart failure with preserved ejection fraction (HFpEF) and heart failure with reduced ejection fraction (HFrEF) [25,26] to have a negative effect on patients’ HRQoL. Evidence also exists, from previous studies, that heart failure is associated with worse quality of life compared to other chronic diseases [27].

In an analysis of the GCOR data base, the presence of peripheral vascular disease strongly correlated with worse quality of life at baseline. The presence of peripheral vascular disease is well established in the literature to be associated with a poor quality of life, affecting quality of life on all domains [28]. The CADANCE study furthermore confirmed that if symptomatic peripheral vascular disease and coronary artery disease coexist, patients will have a considerably higher burden of angina and reduced quality of life [29].

Analysis of the GCOR registry identified that CKD was strongly associated with poor self-assessment of quality of life at baseline. This was in line with the literature establishing a very strong association between different stages of chronic kidney disease (CKD) and HRQoL, with end-stage renal disease having the biggest impact on poor HRQoL [30,31]. The strongest association between CKD and HRQoL was documented in female patients, patients with diabetes, and the presence of cardiovascular and cerebrovascular co-morbidities [30]. In CKD patients a very close relationship has also been established between HRQoL and cardiovascular mortality [31,32].

### 4.2. HRQoL and Association with Outcome (Mortality, MACE, Unplanned Readmission, Adherence to Medication, and Attendance of Cardiac Rehabilitation)

In patients with a diagnosis of coronary artery disease and treated with percutaneous intervention, evidence suggests that baseline health status of patients as measured with the EQ-5D-5L instrument is significantly associated with mortality independent of the clinical variables and complexity of disease [9]. The strongest predictors were functional impairment as well as very poor self-assessment of health status by patients, while the emotional domains did not have a major effect on mortality [14]. From the descriptive components of the EQ-5D-5L, the domains of personal care and mobility were the strongest independent predictors of mortality [12,15], while anxiety/depression did not really contribute to an increased risk of mortality. In this study similar outcomes were noted, confirming both mobility and personal care to be significant independent contributors to the increased risk of mortality. Involvement of all five domains was associated with an increased rate of death compared with patients with no domains affected. When patients reported very poor health (VAS ≤ 60), the adjusted risk of death was 2.4 times higher compared to patients who perceived their health as good (VAS > 60). These results were in concordance with multiple other studies confirming a significant 2–3 times increase in all-cause mortality at 1 year in patients with established coronary artery disease and a very poor baseline QoL [12,15]. A very strong temporal association between poor HRQoL at baseline, mortality and readmissions also exist, and has been demonstrated in a Danish study with significantly increased 5-year mortality and cardiac readmissions [4].

Unplanned readmissions after percutaneous intervention (PCI) for coronary artery disease is a very important quality metric, given its impact on the patient and the cost of healthcare [14]. Very poor quality of life has been demonstrated to be strongly associated with an increased rate of readmissions after PCI [5,6], not only short term (30-day readmission rate) [33] but also long-term. In the TRANSLATE-ACS study, the strongest predictor of 30-day unplanned readmission was poor quality of life, independent of the presenting diagnosis, procedural success, or co-morbidities of patients [33]. This associated risk for unplanned readmissions in patients with poor quality of life was also evident over a longer follow-up period of 3 to 5 years, suggesting a persistent long-term risk [4,14]. In this study, looking at patients from the GCOR database, all the modalities of the EQ-5D-5L, except for personal care and anxiety/depression were associated with an increased risk of unplanned cardiac readmissions. Patients with poor self-reported quality of life (VAS) had a higher risk of unplanned cardiac readmissions compared with patients with good self-reported quality of life but did not reach clinical significance. The assessment of QoL at baseline has the potential of identifying patients at increased risk of unplanned cardiac readmissions with the potential of reducing cost significantly.

In the GCOR study, quality of life did not demonstrate any significant association between either adherence to secondary prevention medication or attendance of cardiac rehabilitation. Very limited information and conflicting results are available in the literature on the possible association between HRQoL at baseline and adherence to secondary prevention medication in patients with cardiovascular disease and after PCI.

### 4.3. HRQoL and Association with Sex

In the literature, women persistently reported worse health related quality of life, independent from the method of assessment [34,35]. The lower health status in women was unrelated to clinical presentation or severity of coronary artery disease [8]. This was true for all women from different population groups and did not relate to an increased rate in mortality [36]. In the VIRGO study (Variation in Recovery: Role of Gender on Outcomes of Young AMI Patients), women reported a significantly reduced quality of life compared to men regardless of pre-existing CAD or not [35]. Furthermore, this significant differences in HRQoL between women and men even persisted long-term after the index event [6]. In the GCOR registry women reported significantly worse quality of life based on all five dimensions of the descriptive EQ-5D-5L patient survey as well as on the visual analogue scale (67.7 in women vs. 71.7 in men, *p* < 0.001). This occurred despite having a lower burden of coronary disease, suggesting that other factors unrelated to disease severity might be implicated. Baseline QoL was significantly associated with subsequent MACE and mortality in the whole cohort. This association was observed for men and women, however, was not statistically significant in women, likely due to a smaller number of women. In the literature a marked difference between men and women were noted for the determining factors related to poor HRQoL, with studies suggesting social and psychological factors to play a more significant role in women. Some studies allude to the fact that gender roles might have an even greater impact on the differences seen comparing women and men [18,34,36]. In the GENESIS-PRAXY study (GENdEr and Sex determInantS of cardiovascular disease: from bench to beyond Premature Acute Coronary Syndrome) no differences in cardiovascular event rates were noted between women and men, but when adjusted for gender traits, a significant increase was noted in patients testing more positive for feminine personality characteristics [37,38]. The lack of biological factors only, to explain the persistent differences in self-reported quality of life between women and men, were further tested in a study by Norris and co-workers. They were able to demonstrate that, when incorporating a gender index (GI), the significant differences between women and men in self-reported quality of life were to a large extent attenuated [34].

It is clear from this study, and comprehensively supported from the literature [39], that both sex (biological) and gender differences should be considered when assessing for differences in quality of life between women and men. The confusing factor from the literature is the interchangeable use of sex and gender, and not clearly adjusting for the differences.

### 4.4. Association between Age and HRQoL

When patients were assessed in the GCOR registry for an association between age and HRQoL, older patients experienced a significant impairment of mobility, personal care, and usual activity, while pain/discomfort was not significantly impaired compared to the younger group of patients and anxiety/depression was significantly less affected in the older patient cohort. Conversely with the VAS score, older patients perceived their general state of health significantly better than the younger patients. These findings from the GCOR database were in line with other studies, including the RITA-2 trial [40] where aging was independently associated with higher health related quality of life in spite of an overall decline in physical functioning and mobility. Similar findings were noted from the PREMIER registry where older patients had a better baseline HRQoL, with this trend persisting for the 12-month follow-up period in spite of a higher mortality rate in older patients [41]. A possible explanation for this better status of health witnessed in the elderly, despite more functional disability and higher mortality, might be that older patients are more complacent regarding their health status and more accepting of the level of functional impairment at their age. They may also require smaller adjustments in the activities of daily living, especially as they are less likely to be employed or have dependents requiring their support.

## 5. Limitations

Potential limitations were observed in this study. By including only patients undergoing PCI, the study was not representative of all patients with coronary artery disease. A further possible limitation to the study was that a generic instrument, and not a disease specific instrument was used to assess quality of life, and the specific impact of angina symptoms on quality of life could not be assessed. The EQ-5D instrument however is well established and validated in the assessment of quality of life in patients with coronary artery disease. Another possible limitation was that the study looked only at sex (biological) differences and did not adjust for sex-gender interaction and its possible impact on quality of life.

## 6. Conclusions

This study confirmed that HRQoL is associated with subsequent MACE and mortality in patients undergoing PCI. By not assessing quality of life as standard of care, we are missing an opportunity to identify high-risk patients who may benefit from targeted interventions to improve health outcomes. This could include reducing unplanned readmissions and henceforth healthcare costs. Furthermore, the significant differences noted between women and men in baseline quality of life were unexplained by measures of disease severity, and further research is needed to address the differences.

## Figures and Tables

**Figure 1 jcm-11-05231-f001:**
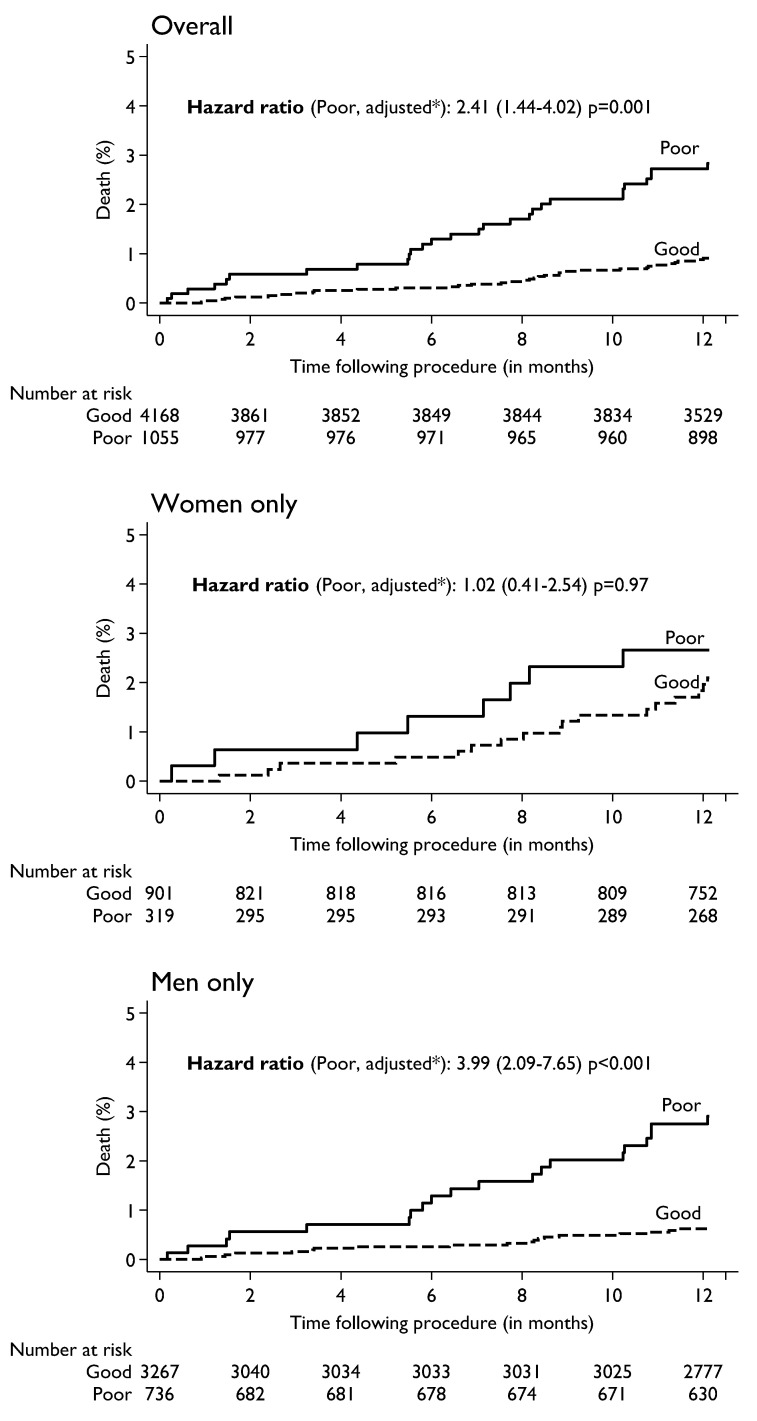
Pre-procedure HRQoL (N = 5223) and its association with mortality within one-year following procedure in the Overall (top), women only (middle) and men only (bottom panel). * Adjusted for age, sex (except for gender specific analysis), BMI, smoking, PCI presentation, diabetes, LVEF, previous MI, PVD, previous CABG, HF within prior 2 weeks of PCI, renal failure and cardiogenic shock.

**Figure 2 jcm-11-05231-f002:**
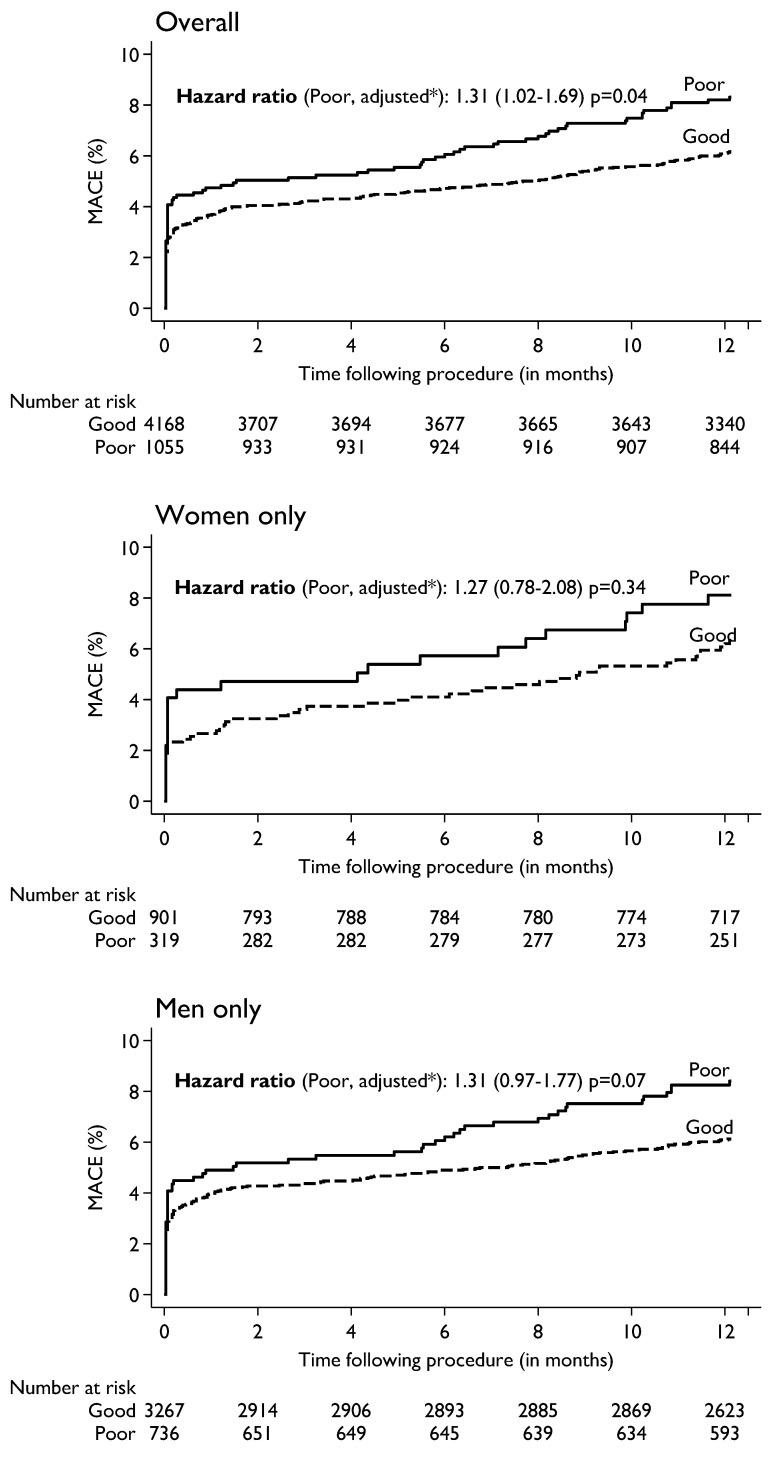
Pre-procedure HRQoL (N = 5223) and its association with MACE within one-year following procedure in the Overall (top), women only (middle) and men only (bottom panel). MACE = major adverse cardiovascular events. * Adjusted for age, sex (except for gender specific analysis), BMI, smoking, PCI presentation, diabetes, LVEF, previous MI, PVD, previous CABG, HF within prior 2 weeks of PCI, renal failure and cardiogenic shock.

**Table 1 jcm-11-05231-t001:** Baseline demographics and clinical presentation of patients by good and poor QoL based on VAS score.

Characteristics n (%)	Total	Good QoL (60+)	Poor QoL (<60)	*p*-Value
N	5223	4168	1055	
QoL score, mean (SD)	70.7 (19.9) (n = 5223)	78.8 (10.8) (n = 4168)	38.8 (14.8) (n = 1055)	<0.001
Male	4003 (76.6%)	3267 (78.4%)	736 (69.8%)	<0.001
Age, years, mean (SD)	68.9 (10.0) (n = 5223)	69.2 (9.7) (n = 4168)	68.0 (10.9) (n = 1055)	<0.001
Diabetes	1240 (23.7%)	956 (22.9%)	284 (26.9%)	0.01
Hypertension	3764 (72.1%)	2996 (71.9%)	768 (72.8%)	0.55
Family History of CAD	1727 (33.1%)	1388 (33.3%)	339 (32.1%)	0.47
Smoking (past or current)	2840 (54.4%)	2244 (53.8%)	596 (56.5%)	0.12
BMI, kg/m^2^ ± SD	29.0 (5.0) (n = 5223)	28.8 (4.8) (n = 4168)	29.6 (5.9) (n = 1055)	<0.001
LVEF, mean ± SD	56.8 (9.4) (n = 5223)	57.1 (9.2) (n = 4168)	55.3 (10.0) (n = 1055)	<0.001
Previous MI	1130 (21.6%)	871 (20.9%)	259 (24.5%)	0.01
Previous Peripheral Vascular Disease	363 (7.0%)	265 (6.4%)	98 (9.3%)	<0.001
Previous PCI	1565 (30.0%)	1242 (29.8%)	323 (30.6%)	0.60
Previous Cerebrovascular disease	368 (7.0%)	283 (6.8%)	85 (8.1%)	0.15
Previous CABG	528 (10.1%)	395 (9.5%)	133 (12.6%)	0.003
Previous HF	274 (5.2%)	206 (4.9%)	68 (6.4%)	0.05
Current HF (<2 wks)	157 (3.0%)	105 (2.5%)	52 (4.9%)	<0.001
Renal failure ^1^	230 (4.4%)	159 (3.8%)	71 (6.7%)	<0.001
Clinical Presentation				
STEMI	307 (5.9%)	199 (4.8%)	108 (10.2%)	<0.001
NSTEMI	1132 (21.7%)	826 (19.8%)	306 (29.0%)	<0.001
Unstable angina	851 (16.3%)	672 (16.1%)	179 (17.0%)	0.51
Elective	2933 (56.2%)	2471 (59.3%)	462 (43.8%)	<0.001
Cardiogenic Shock	9 (0.2%)	4 (0.1%)	5 (0.5%)	0.01
Disease extent-Multivessel	2118 (40.6%)	1708 (41.0%)	410 (38.9%)	0.21

CAD-coronary artery disease; BMI-Body mass index; LVEF-Left ventricular ejection fraction; CABG-Coronary artery bypass grafting; MI-Myocardial infarction; HF-Heart failure; STEMI-ST-elevation myocardial infarction; NSTEMI-Non-ST-elevated myocardial infarction; SD-Standard deviation. ^1^ Renal failure/impairment is defined as either (a) Sr. Creatinine >2 mg/dL and/or (b) having renal failure/receiving dialysis.

**Table 2 jcm-11-05231-t002:** Assessing QoL and VAS for difference between men and women.

Factor	Total	Women	Men	*p*-Value
	5223	1220	4003	
**Mobility**				<0.001
No problem	3740 (71.8%)	771 (63.3%)	2969 (74.3%)	
Some problem	247 (4.7%)	77 (6.3%)	170 (4.3%)	
Moderate problem	1064 (20.4%)	318 (26.1%)	746 (18.7%)	
Severe problem	50 (1.0%)	19 (1.6%)	31 (0.8%)	
Unable/Extreme problem	111 (2.1%)	33 (2.7%)	78 (2.0%)	
**Personal Care**				0.002
No problem	4662 (89.6%)	1051 (86.5%)	3611 (90.5%)	
Some problem	123 (2.4%)	41 (3.4%)	82 (2.1%)	
Moderate problem	336 (6.5%)	95 (7.8%)	241 (6.0%)	
Severe problem	12 (0.2%)	4 (0.3%)	8 (0.2%)	
Unable/Extreme problem	73 (1.4%)	24 (2.0%)	49 (1.2%)	
**Usual Activity**				<0.001
No problem	2980 (57.2%)	610 (50.2%)	2370 (59.4%)	
Some problem	350 (6.7%)	105 (8.6%)	245 (6.1%)	
Moderate problem	1462 (28.1%)	379 (31.2%)	1083 (27.1%)	
Severe problem	63 (1.2%)	23 (1.9%)	40 (1.0%)	
Unable/Extreme problem	352 (6.8%)	97 (8.0%)	255 (6.4%)	
**Pain/Discomfort**				<0.001
No problem	2703 (52.0%)	553 (45.6%)	2150 (53.9%)	
Some problem	432 (8.3%)	118 (9.7%)	314 (7.9%)	
Moderate problem	1680 (32.3%)	416 (34.3%)	1264 (31.7%)	
Severe problem	79 (1.5%)	27 (2.2%)	52 (1.3%)	
Unable/Extreme problem	309 (5.9%)	99 (8.2%)	210 (5.3%)	
**Anxiety/Depression**				<0.001
No problem	3322 (63.8%)	661 (54.4%)	2661 (66.7%)	
Some problem	392 (7.5%)	127 (10.4%)	265 (6.6%)	
Moderate problem	1254 (24.1%)	343 (28.2%)	911 (22.8%)	
Severe problem	34 (0.7%)	11 (0.9%)	23 (0.6%)	
Unable/Extreme problem	204 (3.9%)	74 (6.1%)	130 (3.3%)	
**QoL Score**				
Mean (SD)	70.7 (19.9)	67.7 (21.0)	71.7 (19.4)	<0.001
Median (IQR)	75.0 (60.0, 85.0)	70.0 (55.0, 80.0)	75.0 (60.0, 85.0)	<0.001

SD-Standard deviation; IQR-Interquartile range.

**Table 3 jcm-11-05231-t003:** Assessing QoL descriptive system and VAS for difference by different age groups.

Factor	<55 Years	55–74 Years	≥75 Years	*p*-Value (Overall)	*p*-Value for Trend
	494	3261	1468		
**Mobility**				<0.001	<0.001
No problem	386 (78.5%)	2468 (75.8%)	886 (60.5%)		
Some problem	15 (3.0%)	137 (4.2%)	95 (6.5%)		
Moderate problem	72 (14.6%)	557 (17.1%)	435 (29.7%)		
Severe problem	2 (0.4%)	21 (0.6%)	27 (1.8%)		
Unable/Extreme problem	17 (3.5%)	73 (2.2%)	21 (1.4%)		
**Personal Care**				<0.001	<0.001
No problem	449 (91.1%)	2951 (90.8%)	1262 (86.2%)		
Some problem	9 (1.8%)	63 (1.9%)	51 (3.5%)		
Moderate problem	21 (4.3%)	182 (5.6%)	133 (9.1%)		
Severe problem	2 (0.4%)	7 (0.2%)	3 (0.2%)		
Unable/Extreme problem	12 (2.4%)	46 (1.4%)	15 (1.0%)		
**Usual Activity**				<0.001	0.01
No problem	274 (55.6%)	1959 (60.3%)	747 (51.0%)		
Some problem	21 (4.3%)	196 (6.0%)	133 (9.1%)		
Moderate problem	144 (29.2%)	834 (25.7%)	484 (33.1%)		
Severe problem	3 (0.6%)	36 (1.1%)	24 (1.6%)		
Unable/Extreme problem	51 (10.3%)	225 (6.9%)	76 (5.2%)		
**Pain/Discomfort**				<0.001	0.19
No problem	232 (47.1%)	1792 (55.1%)	679 (46.6%)		
Some problem	32 (6.5%)	252 (7.7%)	148 (10.2%)		
Moderate problem	174 (35.3%)	975 (30.0%)	531 (36.4%)		
Severe problem	10 (2.0%)	38 (1.2%)	31 (2.1%)		
Unable/Extreme problem	45 (9.1%)	195 (6.0%)	69 (4.7%)		
**Anxiety/Depression**				<0.001	0.01
No problem	291 (59.0%)	2087 (64.2%)	944 (64.6%)		
Some problem	31 (6.3%)	229 (7.0%)	132 (9.0%)		
Moderate problem	125 (25.4%)	789 (24.3%)	340 (23.3%)		
Severe problem	3 (0.6%)	22 (0.7%)	9 (0.6%)		
Unable/Extreme problem	43 (8.7%)	125 (3.8%)	36 (2.5%)		
**QoL Score**					
Mean (SD)	66.7 (21.8)	71.5 (20.0)	70.4 (18.7)	<0.001	0.54
Median (IQR)	70.0 (50.0, 80.0)	75.0 (60.0, 85.0)	75.0 (60.0, 80.0)	<0.001	

SD-Standard deviation; IQR-Interquartile range.

**Table 4 jcm-11-05231-t004:** Outcome (following procedure) by presence of problem among those with QoL information available and completed 1-year Follow-up.

Characteristics n (%)	No Problem	With Any Problem	*p*-Value
**Mobility**	**3543**	**1376**	
Death	31 (0.9%)	25 (1.8%)	0.01
MACE	201 (5.7%)	118 (8.6%)	<0.001
MI	130 (3.7%)	85 (6.2%)	<0.001
TVR	60 (1.7%)	25 (1.8%)	0.77
Unplanned Cardiac readmission	312 (8.8%)	176 (12.8%)	<0.001
**Personal Care**	**4413**	**506**	
Death	46 (1.0%)	10 (2.0%)	0.06
MACE	280 (6.3%)	39 (7.7%)	0.24
MI	186 (4.2%)	29 (5.7%)	0.11
TVR	80 (1.8%)	5 (1.0%)	0.18
Unplanned Cardiac readmission	429 (9.7%)	59 (11.7%)	0.17
**Usual Activity**	**2833**	**2086**	
Death	27 (1.0%)	29 (1.4%)	0.15
MACE	165 (5.8%)	154 (7.4%)	0.03
MI	103 (3.6%)	112 (5.4%)	0.003
TVR	52 (1.8%)	33 (1.6%)	0.50
Unplanned Cardiac readmission	243 (8.6%)	245 (11.7%)	<0.001
**Pain/Discomfort**	**2569**	**2350**	
Death	26 (1.0%)	30 (1.3%)	0.38
MACE	144 (5.6%)	175 (7.4%)	0.01
MI	92 (3.6%)	123 (5.2%)	0.01
TVR	41 (1.6%)	44 (1.9%)	0.46
Unplanned Cardiac readmission	226 (8.8%)	262 (11.1%)	0.01
**Anxiety/Depression**	**3143**	**1776**	
Death	36 (1.1%)	20 (1.1%)	0.95
MACE	215 (6.8%)	104 (5.9%)	0.18
MI	140 (4.5%)	75 (4.2%)	0.70
TVR	60 (1.9%)	25 (1.4%)	0.19
Unplanned Cardiac readmission	312 (9.9%)	176 (9.9%)	0.98
**Presence of all 5 Problem**	**No Problem, 1561**	**All 5D Problems, 271**	
Death	16 (1.0%)	7 (2.6%)	0.03
MACE	87 (5.6%)	22 (8.1%)	0.10
MI	52 (3.3%)	15 (5.5%)	0.07
TVR	25 (1.6%)	3 (1.1%)	0.54
Unplanned Cardiac readmission	132 (8.5%)	31 (11.4%)	0.11

MI—Myocardial infarction; TVR—Target vessel revascularization; MACE—Major adverse cardiovascular events.

**Table 5 jcm-11-05231-t005:** Outcome (following discharge) of patients with completed VAS score and completed 1-year Follow-up.

Characteristics	Good QoL (VAS ≥60)	Poor QoL (VAS <60)	*p*-Value
**Overall**	**3923**	**996**	
Death	0.8% (32/3923)	2.4% (24/996)	<0.001
MACE	6.1% (240/3923)	7.9% (79/996)	0.04
MI	4.1% (160/3923)	5.5% (55/996)	0.047
TVR	1.9% (74/3923)	1.1% (11/996)	0.09
Unplanned cardiac readmission	9.6% (377/3923)	11.1% (111/996)	0.15
Rehab attendance	46.5% (1657/3563)	46.4% (423/912)	0.95
Use of all 4 drugs * at 1 yr	26.3% (967/3677)	28.0% (255/912)	0.31
**Among Female**	**832**	**300**	
Death	1.8% (15/832)	2.3% (7/300)	0.57
MACE	6.3% (52/832)	8.0% (24/300)	0.30
MI	3.8% (32/832)	5.7% (17/300)	0.18
TVR	1.6% (13/832)	0.7% (2/300)	0.24
Unplanned cardiac readmission	10.5% (87/832)	11.0% (33/300)	0.79
Rehab attendance	46.8% (355/758)	38.4% (106/276)	0.02
Use of all 4 drugs * at 1 yr	23.7% (185/782)	26.6% (73/274)	0.32
**Among Male**	**3091**	**696**	
Death	0.5% (17/3091)	2.4% (17/696)	<0.001
MACE	6.1% (188/3091)	7.9% (55/696)	0.08
MI	4.1% (128/3091)	5.5% (38/696)	0.12
TVR	2.0% (61/3091)	1.3% (9/696)	0.23
Unplanned cardiac readmission	9.4% (290/3091)	11.2% (78/696)	0.14
Rehab attendance	46.4% (1302/2805)	49.8% (317/636)	0.12
Use of all 4 drugs * at 1 yr	27.0% (782/2895)	28.5% (182/638)	0.44

* Angiotensin-converting enzyme inhibitors/angiotensin II receptor blockers + betablockers + statin + any Dual antiplatelet therapy. MI—Myocardial infarction; TVR—Target vessel revascularization; MACE—Major adverse cardiovascular events.

## Data Availability

The data underlying this article will be shared on reasonable request to the corresponding author.

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
