# Peer review of "Health-Related Quality of Life (HRQoL) and the Effect on Outcome in Patients Presenting with Coronary Artery Disease and Treated with Percutaneous Coronary Intervention (PCI): Differences Noted by Sex and Age"

_jcm, 2022, doi:10.3390/jcm11175231_

Round 1

Reviewer 1 Report

I have read with interest this study exploring the impact of Healy-related quality of life on outcomes after PCI. Overall, the paper is well presented. Methods and statistical analyses should be improved. Here below you can find my comments.

  • The introduction might be shortened. Most of it may be used for the discussion.
  • the Title as well as the conclusions on “marked difference” by sex are unjustified by the analyses that are underpowered by the limited sample size (females)
  • Similarly, the conclusions on sex-gender interaction is speculative and should be kept in the discussion only.
  • Statistical Analyses need a major revision. Important differences are shown between groups (QoL) and all the results might be driven by these confounding factors. I would suggest providing univariable and multivariable cox regression analyses including more factors than those used in the current form. 
  • Please correct Table 5 title (Table 1) 
  • Typo line 111
  • All the analyses are performed on patients with data on EQ-5D available. Please avoid repeating it in all the manuscript paragraphs.
  • Line 197. Adjustment for age is not shown.

Reviewer 2 Report

The manuscript, especially the discussion, is very interesting and well-written, though the study is not innovative. The cohort is large (15,198 patients).

There are several issues that need to be addressed:

1.     Only 6591 (43.4%) completed the self-assessment. Why is participation so low? There exists a selection bias.

2.     Lack of data on coronary angiography and procedure. 1, 2 or 3 vessel disease? How many patients undergoing complete revascularization?  These will influence HRQoL.

3.     Table 5? Two table 1?

Reviewer 3 Report

Major comments

A total of number of cohort in Figure 1 is 6581 patients (5283 in Good and 1298 in Poor). I guess that it should be 6591 same as that in Table 3, please explain.

Furthermore, I can’t understand the total number of 6570 patients in Figure 2 (5274 in Good and 1296 in Poor).

Why the total number of patients are different between Table 4 and 5?

P9, L255; Authors stated “Kaplan-Meier curves and adjusted Cox hazard models (Figure 1)”. Was the curves drawn in Figure 1 K-M curves or cumulative hazard risk from Cox regression model? If it is K-M curve, patients without data on age or BMI are regarded as including into these K-M curves. But these patients without data on age or BMI are regarded as excluding from HR 2.85 (1.76-4.62) stated in Figure 1, since cox hazard models can’t treat them. Hence, K-M curve and HR in Figure 1 were calculated from different patients' population. That can also be said about other Figures.

This manuscript is hard to understand since all of Tables and Figures assess the different population. Authors should simplify the study population. For example, I suggest to exclude the patients with a defect of data (ex. age, BMI) and/or use a patients’ inclusion chart to help the understand the patients flow.

The aim of this study is to assess the correlation between the HRQoL and clinical outcomes, Hence Table 1 is not required because of no relevance to the aim of this study. 

A total of number of cohort in Figure 1 is 6581 patients (5283 in Good and 1298 in Poor). I guess that it should be 6591 same as that in Table 3, please explain.

Furthermore, I can’t understand the total number of 6570 in Figure 2 (5274 in Good and 1296 in Poor).

The description in "3.2 Baseline characteristics of patients by self-assessed HRQoL status and differences observed 162 for sex and age" is the repeat of result of Table 2. You should avoid to repeat numbers in the text since redundancy writting.

Minor comments

Introduction section is lengthy. Detail on EQ-5D-5L should be stated in Methods section.

Please define the renal failure in Table 1 and 2.

Please revise the title of Table 5; Table 1 Table 5

Lack of total number in left column in Table 2.

Please unify the number of decimal place in P value column in Tables.

Round 2

Reviewer 1 Report

The paper is improved. Comments have been adequately addressed.

Reviewer 2 Report

I have no other comments.

Reviewer 3 Report

My concerns were corrected properly.